# Provably Efficient Causal Reinforcement Learning with Confounded Observational Data

**Lingxiao Wang**
Northwestern University
lwang@u.northwestern.edu

**Zhuoran Yang**
Princeton University
zy6@princeton.edu

**Zhaoran Wang**
Northwestern University
zhaoranwang@gmail.com

## Abstract

Empowered by neural networks, deep reinforcement learning (DRL) achieves tremendous empirical success. However, DRL requires a large dataset by interacting with the environment, which is unrealistic in critical scenarios such as autonomous driving and personalized medicine. In this paper, we study how to incorporate the dataset collected in the offline setting to improve the sample efficiency in the online setting. To incorporate the observational data, we face two challenges. (a) The behavior policy that generates the observational data may depend on unobserved random variables (confounders), which affect the received rewards and transition dynamics. (b) Exploration in the online setting requires quantifying the uncertainty given both the observational and interventional data. To tackle such challenges, we propose the deconfounded optimistic value iteration (DOVI) algorithm, which incorporates the confounded observational data in a provably efficient manner. DOVI explicitly adjusts for the confounding bias in the observational data, where the confounders are partially observed or unobserved. In both cases, such adjustments allow us to construct the bonus based on a notion of information gain, which takes into account the amount of information acquired from the offline setting. In particular, we prove that the regret of DOVI is smaller than the optimal regret achievable in the pure online setting when the confounded observational data are informative upon the adjustments.

## 1 Introduction

Empowered by the breakthrough in neural networks, deep reinforcement learning (DRL) achieves significant empirical successes in various scenarios [19, 23, 36, 37]. Learning an expressive function approximator necessitates collecting a large dataset. Specifically, in the online setting, it requires the agent to interact with the environment for a large number of steps. For example, to learn a human-level policy for playing Atari games, the agent has to interact with a simulator for more than $10^8$ steps [13]. However, in most scenarios, we do not have access to a simulator that allows for trial and error without any cost. Meanwhile, in critical scenarios, e.g., autonomous driving and personalized medicine, trial and error in the real world is unsafe and even unethical. As a result, it remains challenging to apply DRL to more scenarios.

To bypass such a barrier, we study how to incorporate the dataset collected offline, namely the observational data, to improve the sample efficiency of RL in the online setting [21]. In contrast to the interventional data collected online in possibly expensive ways, observational data are often abundantly available in various scenarios. For example, in autonomous driving, we have access to trajectories generated by the drivers. As another example, in personalized medicine, we have access to electronic health records from doctors. However, to incorporate the observational data in a provably efficient way, we have to address two challenges.

35th Conference on Neural Information Processing Systems (NeurIPS 2021).

- The observational data are possibly confounded. Specifically, there often exist unobserved random variables, namely confounders, that causally affect the agent and the environment at the same time. In particular, the policy used to generate the observational data, namely the behavior policy, possibly depends on the confounders. Meanwhile, the confounders possibly affect the received rewards and the transition dynamics.

  In the example of autonomous driving [9, 22], the drivers may be affected by complicated traffic or poor road design, resulting in traffic accidents even without misconduct. The complicated traffic and poor road design subsequently affect both the action of the drivers and the outcome. Therefore, it is unclear from the observational data whether the accidents are due to the actions adopted by the drivers. Agents trained with such observational data may be unwilling to take any actions under complicated traffic, jeopardizing the safety of passengers.

  In the example of personalized medicine [8, 29], the patients may not be compliant with prescriptions and instructions, which subsequently affects both the treatment and the outcome. As another example, the doctor may prescribe medicine to patients based on patients' socioeconomic status (which could be inferred by the doctor through interacting with the patients). Meanwhile, socioeconomic status affects the patients' health condition and subsequently plays the role of the confounder. In both scenarios, such confounders may be unavailable due to privacy or ethical concerns. Such a confounding issue makes the observational data uninformative and even misleading for identifying and estimating the causal effect, which is crucial for decision-making in the online setting. In all the examples, it is unclear from the observational data whether the outcome is due to the actions adopted.

- Even without the confounding issue, it remains unclear how the observational data may facilitate exploration in the online setting, which is the key to the sample efficiency of RL. At the core of exploration is uncertainty quantification. Specifically, quantifying the uncertainty that remains given the dataset collected up to the current step, including the observational data and the interventional data, allows us to construct a bonus. When incorporated into the reward, such a bonus encourages the agent to explore the less visited state-action pairs with more uncertainty. In particular, constructing such a bonus requires quantifying the amount of information carried over by the observational data from the offline setting, which also plays a key role in characterizing the regret, especially how much the observational data may facilitate reducing the regret.

  Uncertainty quantification becomes even more challenging when the observational data are confounded. Specifically, as the behavior policy depends on the confounders, there is a mismatch between the data generating processes in the offline setting and the online setting. As a result, it remains challenging to quantify how much information carried over from the offline setting is useful for the online setting, as the observational data are uninformative and even misleading due to the confounding issue.

**Contribution.** To study causal reinforcement learning, we propose a class of Markov decision processes (MDPs), namely confounded MDPs, which captures the data generating processes in both the offline setting and the online setting as well as their mismatch due to the confounding issue. In particular, we study two tractable cases of confounded MDPs in the episodic setting with linear function approximation [7, 16, 42, 43].

- In the first case, the confounders are partially observed in the observational data. Assuming that an observed subset of the confounders satisfies the backdoor criterion [32], we propose the deconfounded optimistic value iteration (DOVI) algorithm, which explicitly corrects for the confounding bias in the observational data using the backdoor adjustment.

- In the second case, the confounders are unobserved in the observational data. Assuming that there exists an observed set of intermediate states that satisfies the frontdoor criterion [32], we propose an extension of DOVI, namely DOVI$^+$, which explicitly corrects for the confounding bias in the observational data using the composition of two backdoor adjustments. We remark that DOVI$^+$ follows the same principle of design as DOVI and defer the discussion of DOVI$^+$ to §A.

In both cases, the adjustments allow DOVI and DOVI$^+$ to incorporate the observational data into the interventional data while bypassing the confounding issue. It further enables estimating the causal effect of a policy on the received rewards and the transition dynamics with enlarged effective sample size. Moreover, such adjustments allow us to construct the bonus based on a notion of information gain, which takes into account the amount of information carried over from the offline setting.

In particular, we prove that DOVI and DOVI$^+$ attain the $\Delta_H \cdot \sqrt{d^3 H^3 T}$-regret up to logarithmic factors, where $d$ is the dimension of features, $H$ is the length of each episode, and $T = HK$ is the number of steps taken in the online setting, where $K$ is the number of episodes. Here the multiplicative factor $\Delta_H > 0$ depends on $d$, $H$, and a notion of information gain that quantifies the amount of information obtained from the interventional data additionally when given the properly adjusted observational data. When the observational data are unavailable or uninformative upon the adjustments, $\Delta_H$ is a logarithmic factor. Correspondingly, DOVI and DOVI$^+$ attain the optimal $\sqrt{T}$-regret achievable in the pure online setting [7, 16, 42, 43]. When the observational data are sufficiently informative upon the adjustments, $\Delta_H$ decreases towards zero as the effective sample size of the observational data increases, which quantifies how much the observational data may facilitate exploration in the online setting.

**Related Work.** Our work is related to the study of causal bandit [20]. The goal of causal bandit is to obtain the optimal intervention in the online setting where the data generating process is described by a causal diagram. The previous study establishes causal bandit algorithms in the online setting [26, 34], the offline setting [17, 18], and a combination of both settings [11]. In contrast to this line of work, we study causal RL in a combination of the online setting and the offline setting. Causal RL is more challenging than causal bandit, which corresponds to $H = 1$, as it involves the transition dynamics and is more challenging in exploration. See §B for a detailed literature review on causal bandit.

Our work is related to the study of causal RL considered in various settings. [45] propose a model-based RL algorithm that solves dynamic treatment regimes (DTR), which involve a combination of the online setting and the offline setting. Their algorithm hinges on the analysis of sensitivity [3, 27, 38, 44], which constructs a set of feasible models of the transition dynamics based on the confounded observational data. Correspondingly, their algorithm achieves exploration by choosing an optimistic model of the transition dynamics from such a feasible set. In contrast, we propose a model-free RL algorithm, which achieves exploration through the bonus based on a notion of information gain. It is worth mentioning that the assumption of [45] is weaker than ours as theirs does not allow for identifying the causal effect. As a result of partial identification, the regret of their algorithm is the same as the regret in the pure online setting as $T \to +\infty$. In contrast, our work instantiates the following framework in handling confounders for reinforcement learning. (a) First, we propose the estimation equation based on the observations, which identifies the causal effect of actions on the cumulative reward. (b) Second, we conduct point estimation and uncertainty quantification based on observations and the estimation equation. (c) Finally, we conduct exploration based on the uncertainty quantification and achieve the regret reduction in the online setting. Consequently, the regret of our algorithm is smaller than the regret in the pure online setting by a multiplicative factor for all $T$. [25] propose a model-based RL algorithm in a combination of the online setting and the offline setting. Their algorithm uses a variational autoencoder (VAE) for estimating a structural causal model (SCM) based on the confounded observational data. In particular, their algorithm utilizes the actor-critic algorithm to obtain the optimal policy in such an SCM. However, the regret of their algorithm remains unclear. [6] propose a model-based RL algorithm in the pure online setting that learns the optimal policy in a partially observable Markov decision process (POMDP). The regret of their algorithm also remains unclear. [35] utilize generative adversarial reinforcement learning to reconstruct transition dynamics with confounder, and [40] propose a model-based approach for POMDP based on adjustment with proxy variables. [30] consider off-policy policy evaluation under one-decision confounding and constructs worst-case bounds with theoretical guarantee. [4] utilizes states and actions as proxy variables to tackle off-policy policy evaluation with confounders. In contrast, our work utilizes backdoor and frontdoor adjustments to handle confounded observation.

## 2   Confounded Reinforcement Learning

**Structural Causal Model.** We denote a structural causal model (SCM) [32] by a tuple $(A, B, F, P)$. Here $A$ is the set of exogenous (unobserved) variables, $B$ is the set of endogenous (observed) variables, $F$ is the set of structural functions capturing the causal relations, which determines an endogenous variable $v \in B$ based on the other exogenous and endogenous variables, and $P$ is the distribution of all the exogenous variables. We say that a pair of variables $Y$ and $Z$ are confounded by a variable $W$ if they are both caused by $W$.

An intervention on a set of endogenous variables $X \subseteq B$ assigns a value $x$ to $X$ regardless of the other exogenous and endogenous variables as well as the structural functions. We denote by $\mathrm{do}(X = x)$ the intervention on $X$ and write $\mathrm{do}(x)$ if it is clear from the context. Similarly, a stochastic intervention [10, 28] on a set of endogenous variables $X \subseteq B$ assigns a distribution $p$ to $X$ regardless of the other exogenous and endogenous variables as well as the structural functions. We denote by $\mathrm{do}(X \sim p)$ the stochastic intervention on $X$.

**Confounded Markov Decision Process.** To characterize a Markov decision process (MDP) in the offline setting with observational data, which are possibly confounded, we introduce an SCM, where the endogenous variables are the states $\{s_h\}_{h \in [H]}$, actions $\{a_h\}_{h \in [H]}$, and rewards $\{r_h\}_{h \in [H]}$. Let $\{w_h\}_{h \in [H]}$ be the confounders. In §3, we assume that the confounders are partially observed, while in §A, we assume that they are unobserved. The set of structural functions $F$ consists of the transition of states $s_{h+1} \sim \mathcal{P}_h(\cdot \,|\, s_h, a_h, w_h)$, the transition of confounders $w_h \sim \widehat{\mathcal{P}}_h(\cdot \,|\, s_h)$, the behavior policy $a_h \sim \nu_h(\cdot \,|\, s_h, w_h)$, which depends on the confounder $w_h$, and the reward function $r_h(s_h, a_h, w_h)$. See Figure 1 for the causal diagram that describes such an SCM.

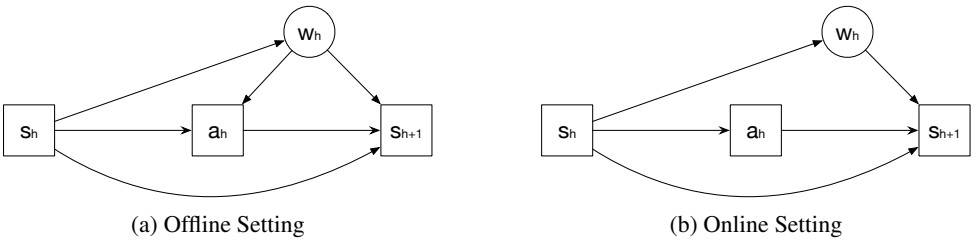

(a) Offline Setting  (b) Online Setting

Figure 1: Causal diagrams of the $h$-th step of the confounded MDP (a) in the offline setting and (b) in the online setting, respectively.

Here $a_h$ and $s_{h+1}$ are confounded by $w_h$ in addition to $s_h$. We denote such a confounded MDP by the tuple $(\mathcal{S}, \mathcal{A}, \mathcal{W}, H, \overline{\mathcal{P}}, r)$, where $H$ is the length of an episode, $\mathcal{S}$, $\mathcal{A}$, and $\mathcal{W}$ are the spaces of states, actions, and confounders, respectively, $r = \{r_h\}_{h \in [H]}$ is the set of reward functions, and $\overline{\mathcal{P}} = \{\mathcal{P}_h, \widetilde{\mathcal{P}}_h\}_{h \in H}$ is the set of transition kernels. In the sequel, we assume without loss of generality that $r_h$ takes value in $[0, 1]$ for all $h \in [H]$.

In the online setting that allows for intervention, we assume that the confounders $\{w_h\}_{h \in [H]}$ are unobserved. A policy $\pi = \{\pi_h\}_{h \in [H]}$ induces the stochastic intervention $\mathrm{do}(a_1 \sim \pi_1(\cdot \,|\, s_1), \ldots, a_H \sim \pi_H(\cdot \,|\, s_H))$, which does not depend on the confounders. In particular, an agent interacts with the environment as follows. At the beginning of the $k$-th episode, the environment arbitrarily selects an initial state $s_1^k$ and the agent selects a policy $\pi^k = \{\pi_h^k\}_{h \in [H]}$. At the $h$-th step of the $k$-th episode, the agent observes the state $s_h^k$ and takes the action $a_h^k \sim \pi_h^k(\cdot \,|\, s_h^k)$. The environment randomly selects the confounder $w_h^k \sim \widetilde{\mathcal{P}}_h(\cdot \,|\, s_h^k)$, which is unobserved, and the agent receives the reward $r_h^k = r_h(s_h^k, a_h^k, w_h^k)$. The environment then transits into the next state $s_{h+1}^k \sim \mathcal{P}_h(\cdot \,|\, s_h^k, a_h^k, w_h^k)$.

For a policy $\pi = \{\pi_h\}_{h \in H}$, which does not depend on the confounders $\{w_h\}_{h \in [H]}$, we define the value function $V^\pi = \{V_h^\pi\}_{h \in [H]}$ as follows,

$$V_h^\pi(s) = \mathbb{E}_\pi \left[ \sum_{j=h}^H r_j(s_j, a_j, w_j) \,\middle|\, s_h = s \right], \quad \forall h \in [H], \tag{2.1}$$

where we denote by $\mathbb{E}_\pi$ the expectation with respect to the confounders $\{w_j\}_{j=h}^H$ and the trajectory $\{(s_j, a_j)\}_{j=h}^H$, starting from the state $s_j = s$ and following the policy $\pi$. Correspondingly, we define the action-value function $Q^\pi = \{Q_h^\pi\}_{h \in [H]}$ as follows,

$$Q_h^\pi(s, a) = \mathbb{E}_\pi \left[ \sum_{j=h}^H r_j(s_j, a_j, w_j) \,\middle|\, s_h = s, \mathrm{do}(a_h = a) \right], \quad \forall h \in [H]. \tag{2.2}$$

We assess the performance of an algorithm using the regret against the globally optimal policy $\pi^* = \{\pi_h^*\}_{h \in [H]}$ in hindsight after $K$ episodes, which is defined as follows,

$$\text{Regret}(T) = \max_{\pi} \sum_{k=1}^{K} \left(V_1^{\pi}(s_1^k) - V_1^{\pi^k}(s_1^k)\right) = \sum_{k=1}^{K} \left(V_1^{\pi^*}(s_1^k) - V_1^{\pi^k}(s_1^k)\right). \tag{2.3}$$

Here $T = HK$ is the total number of steps.

Our goal is to design an algorithm that minimizes the regret defined in (2.3), where $\pi^*$ does not depend on the confounders $\{w_h\}_{h \in [H]}$. In the online setting that allows for intervention, it is well understood how to minimize such a regret [2, 14–16]. However, it remains unclear how to efficiently utilize the observational data obtained in the offline setting, which are possibly confounded. In real-world applications, e.g., autonomous driving and personalized medicine, such observational data are often abundant, whereas intervention in the online setting is often restricted. We refer to §C for a comparison between the confounded MDP and other extensions of MDP, including the dynamics treatment regime (DTR), partially observable MDP (POMDP), and contextual MDP (CMDP).

**Why is Incorporating Confounded Observational Data Challenging?** Straightforwardly incorporating the confounded observational data into an online algorithm possibly leads to an undesirable regret due to the mismatch between the online and offline data generating processes. In particular, due to the existence of the confounders $\{w_h\}_{h \in [H]}$, which are partially observed (§3) or unobserved (§A), the conditional probability $\mathbb{P}(s_{h+1} \mid s_h, a_h)$ in the offline setting is different from the causal effect $\mathbb{P}(s_{h+1} \mid s_h, \text{do}(a_h))$ in the online setting [33]. More specifically, it holds that

$$\mathbb{P}(s_{h+1} \mid s_h, a_h) = \frac{\mathbb{E}_{w_h \sim \widetilde{\mathcal{P}}_h(\cdot \mid s_h)} \left[\mathcal{P}_h(s_{h+1} \mid s_h, a_h, w_h) \cdot \nu_h(a_h \mid s_h, w_h)\right]}{\mathbb{E}_{w_h \sim \widetilde{\mathcal{P}}_h(\cdot \mid s_h)} \left[\nu_h(a_h \mid s_h, w_h)\right]},$$

$$\mathbb{P}(s_{h+1} \mid s_h, \text{do}(a_h)) = \mathbb{E}_{w_h \sim \widetilde{\mathcal{P}}_h(\cdot \mid s_h)} \left[\mathcal{P}_h(\cdot \mid s_h, a_h, w_h)\right].$$

In other words, without proper covariate adjustments [32], the confounded observational data may be not informative for estimating the transition dynamics and the associated action-value function in the online setting. To this end, we propose an algorithm that incorporates the confounded observational data in a provably efficient manner. Moreover, our analysis quantifies the amount of information carried over by the confounded observational data from the offline setting and to what extent it helps reducing the regret in the online setting.

## 3 Algorithm and Theory for Partially Observed Confounder

In this section, we propose the Deconfounded Optimistic Value Iteration (DOVI) algorithm. DOVI handles the case where the confounders are unobserved in the online setting but are partially observed in the offline setting. We then characterize the regret of DOVI. We defer the extension of DOVI, namely DOVI+, to §A which handles the case where the confounders are unobserved in both the online setting and the offline setting.

### 3.1 Algorithm

**Backdoor Adjustment.** In the online setting that allows for intervention, the causal effect of $a_h$ on $s_{h+1}$ given $s_h$, that is, $\mathbb{P}(s_{h+1} \mid s_h, \text{do}(a_h))$, plays a key role in the estimation of the action-value function. Meanwhile, the confounded observational data may not allow us to identify the causal effect $\mathbb{P}(s_{h+1} \mid s_h, \text{do}(a_h))$ if the confounder $w_h$ is unobserved. However, if the confounder $w_h$ is partially observed in the offline setting, the observed subset $u_h$ of $w_h$ allows us to identify the causal effect $\mathbb{P}(s_{h+1} \mid s_h, \text{do}(a_h))$, as long as $u_h$ satisfies the following backdoor criterion.

**Assumption 3.1** (Backdoor Criterion [32, 33]). In the SCM defined in §2 and its induced directed acyclic graph (DAG), for all $h \in [H]$, there exists an observed subset $u_h$ of $w_h$ that satisfies the backdoor criterion, that is,

- the elements of $u_h$ are not the descendants of $a_h$, and

- conditioning on $s_h$, the elements of $u_h$ $d$-separate every path between $a_h$ and $s_{h+1}, r_h$ that has an incoming arrow into $a_h$.

See Figure 2 for an example that satisfies the backdoor criterion. In particular, we identify the causal effect $\mathbb{P}(s_{h+1} \,|\, s_h, \mathrm{do}(a_h))$ as follows.

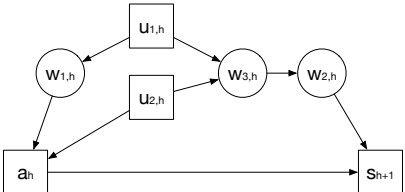

Figure 2: An illustration of the backdoor criterion modified from [32]. The causal diagram corresponds to the $h$-th step of the confounded MDP conditioning on $s_h$. Here $w_h = \{w_{1,h}, w_{2,h}, w_{3,h}\}$ is the unobserved confounders and the subset $u_h = \{u_{1,h}, u_{2,h}\}$ satisfies the backdoor criterion.

**Proposition 3.2** (Backdoor Adjustment [32]). Under Assumption 3.1, it holds for all $h \in [H]$ that

$$\mathbb{P}\big(s_{h+1} \,\big|\, s_h, \mathrm{do}(a_h)\big) = \mathbb{E}_{u_h \sim \mathbb{P}(\cdot \,|\, s_h)}\big[\mathbb{P}(s_{h+1} \,|\, s_h, a_h, u_h)\big],$$

$$\mathbb{E}\big[r_h(s_h, a_h, w_h) \,\big|\, s_h, \mathrm{do}(a_h)\big] = \mathbb{E}_{u_h \sim \mathbb{P}(\cdot \,|\, s_h)}\Big[\mathbb{E}\big[r_h(s_h, a_h, w_h) \,\big|\, s_h, a_h, u_h\big]\Big].$$

Here $(s_{h+1}, s_h, a_h, u_h)$ follows the SCM defined in §2, which generates the confounded observational data.

*Proof.* See [32] for a detailed proof. $\qquad\square$

With a slight abuse of notation, we write $\mathbb{P}(s_{h+1} \,|\, s_h, a_h, u_h)$ as $\mathcal{P}_h(s_{h+1} \,|\, s_h, a_h, u_h)$ and $\mathbb{P}(u_h \,|\, s_h)$ as $\widetilde{\mathcal{P}}_h(u_h \,|\, s_h)$, since they are induced by the SCM defined in §2. In the sequel, we define $\mathcal{U}$ the space of observed state $u_h$ and write $r_h = r_h(s_h, a_h, w_h)$ for notational simplicity.

**Backdoor-Adjusted Bellman Equation.** We now formulate the Bellman equation for the confounded MDP. It holds for all $(s_h, a_h) \in \mathcal{S} \times \mathcal{A}$ that

$$Q_h^\pi(s_h, a_h) = \mathbb{E}_\pi\bigg[\sum_{j=h}^H r_j(s_j, a_j, u_j) \,\bigg|\, s_h, \mathrm{do}(a_h)\bigg] = \mathbb{E}\big[r_h \,\big|\, s_h, \mathrm{do}(a_h)\big] + \mathbb{E}_{s_{h+1}}\big[V_{h+1}^\pi(s_{h+1})\big],$$

where $\mathbb{E}_{s_{h+1}}$ denotes the expectation with respect to $s_{h+1} \sim \mathbb{P}(\cdot \,|\, s_h, \mathrm{do}(a_h))$. Here $\mathbb{E}[r_h \,|\, s_h, \mathrm{do}(a_h)]$ and $\mathbb{P}(\cdot \,|\, s_h, \mathrm{do}(a_h))$ are characterized in Proposition 3.2. In the sequel, we define the following transition operator and counterfactual reward function,

$$(\mathbb{P}_h V)(s_h, a_h) = \mathbb{E}_{s_{h+1} \sim \mathbb{P}(\cdot \,|\, s_h, \mathrm{do}(a_h))}\big[V(s_{h+1})\big], \quad \forall V : \mathcal{S} \mapsto \mathbb{R}, \ (s_h, a_h) \in \mathcal{S} \times \mathcal{A}, \tag{3.1}$$

$$R_h(s_h, a_h) = \mathbb{E}\big[r_h \,\big|\, s_h, \mathrm{do}(a_h)\big], \quad \forall(s_h, a_h) \in \mathcal{S} \times \mathcal{A}. \tag{3.2}$$

We have the following Bellman equation,

$$Q_h^\pi(s_h, a_h) = R_h(s_h, a_h) + (\mathbb{P}_h V_{h+1}^\pi)(s_h, a_h), \quad \forall h \in [H], \ (s_h, a_h) \in \mathcal{S} \times \mathcal{A}. \tag{3.3}$$

Correspondingly, the Bellman optimality equation takes the following form,

$$Q_h^*(s_h, a_h) = R_h(s_h, a_h) + (\mathbb{P}_h V_{h+1}^*)(s_h, a_h), \quad V_h^*(s_h) = \max_{a_h \in \mathcal{A}} Q_h^*(s_h, a_h), \tag{3.4}$$

which holds for all $h \in [H]$ and $(s_h, a_h) \in \mathcal{S} \times \mathcal{A}$. Such a Bellman optimality equation allows us to adapt the least-squares value iteration (LSVI) algorithm [2, 5, 14, 16, 31].

**Linear Function Approximation.** We focus on the following setting with linear transition kernels and reward functions [7, 16, 42, 43], which corresponds to a linear SCM [33].

**Assumption 3.3** (Linear Confounded MDP). We assume that

$$\mathcal{P}_h(s_{h+1} \,|\, s_h, a_h, u_h) = \langle \phi_h(s_h, a_h, u_h), \mu_h(s_{h+1}) \rangle, \quad \forall h \in [H], \ (s_{h+1}, s_h, a_h) \in \mathcal{S} \times \mathcal{S} \times \mathcal{A},$$

where $\phi_h(\cdot, \cdot, \cdot)$ and $\mu_h(\cdot) = (\mu_{1,h}(\cdot), \ldots, \mu_{d,h}(\cdot))^\top$ are $\mathbb{R}^d$-valued functions. We assume that $\sum_{i=1}^d \|\mu_{i,h}\|_1^2 \le d$ and $\|\phi_h(s_h, a_h, u_h)\|_2 \le 1$ for all $h \in [H]$ and $(s_h, a_h, u_h) \in \mathcal{S} \times \mathcal{A} \times \mathcal{U}$. Meanwhile, we assume that

$$\mathbb{E}[r_h \,|\, s_h, a_h, u_h] = \phi_h(s_h, a_h, u_h)^\top \theta_h, \quad \forall h \in [H], \ (s_h, a_h, u_h) \in \mathcal{S} \times \mathcal{A} \times \mathcal{U}, \tag{3.5}$$

where $\theta_h \in \mathbb{R}^d$ and $\|\theta_h\|_2 \le \sqrt{d}$ for all $h \in [H]$.

Such a linear setting generalizes the tabular setting where $\mathcal{S}$, $\mathcal{A}$, and $\mathcal{U}$ are finite.

**Proposition 3.4.** We define the backdoor-adjusted feature as follows,

$$\psi_h(s_h, a_h) = \mathbb{E}_{u_h \sim \widetilde{\mathcal{P}}_h(\cdot \mid s_h)}\big[\phi_h(s_h, a_h, u_h)\big], \quad \forall h \in [H], \ (s_h, a_h) \in \mathcal{S} \times \mathcal{A}. \tag{3.6}$$

Under Assumption 3.1, it holds that

$$\mathbb{P}(s_{h+1} \mid s_h, \mathrm{do}(a_h)) = \langle \psi_h(s_h, a_h), \mu_h(s_{h+1}) \rangle, \quad \forall h \in [H], \ (s_{h+1}, s_h, a_h) \in \mathcal{S} \times \mathcal{S} \times \mathcal{A}.$$

Moreover, the action-value functions $Q_h^\pi$ and $Q_h^*$ are linear in the backdoor-adjusted feature $\psi_h$ for all $\pi$.

*Proof.* See §F.1 for a detailed proof. $\qquad\qquad\qquad\qquad\qquad\qquad\qquad\qquad\qquad\qquad\qquad\square$

Such an observation allows us to estimate the action-value function based on the backdoor-adjusted features $\{\psi_h\}_{h \in [H]}$ in the online setting. See §D for a detailed discussion. In the sequel, we assume that either the density of $\{\widetilde{\mathcal{P}}_h(\cdot \mid s_h)\}_{h \in [H]}$ is known or the backdoor-adjusted feature $\{\psi_h\}_{h \in [H]}$ is known.

In the sequel, we introduce the DOVI algorithm (Algorithm 1). Each iteration of DOVI consists of two components, namely point estimation, where we estimate $Q_h^*$ based on the confounded observational data and the interventional data, and uncertainty quantification, where we construct the upper confidence bound (UCB) of the point estimator.

---

**Algorithm 1** Deconfounded Optimistic Value Iteration (DOVI) for Confounded MDP

---

**Require:** Observational data $\{(s_h^i, a_h^i, u_h^i, r_h^i)\}_{i \in [n], h \in [H]}$, tuning parameters $\lambda, \beta > 0$, backdoor-adjusted feature $\{\psi_h\}_{h \in [H]}$, which is defined in (3.6).
 1: **Initialization:** Set $\{Q_h^0, V_h^0\}_{h \in [H]}$ as zero functions and $V_{H+1}^k$ as a zero function for $k \in [K]$.
 2: **for** $k = 1, \dots, K$ **do**
 3:     **for** $h = H, \dots, 1$ **do**
 4:         Set $\omega_h^k \leftarrow \operatorname{argmin}_{\omega \in \mathbb{R}^d} \sum_{\tau=1}^{k-1}(r_h^\tau + V_{h+1}^\tau(s_{h+1}^\tau) - \omega^\top \psi_h(s_h^\tau, a_h^\tau))^2 + \lambda\|\omega\|_2^2 + L_h^k(\omega)$, where $L_h^k$ is defined in (3.8).
 5:         Set $Q_h^k(\cdot, \cdot) \leftarrow \min\{\psi_h(\cdot, \cdot)^\top \omega_h^k + \Gamma_h^k(\cdot, \cdot), H - h\}$, where $\Gamma_h^k$ is defined in (3.12).
 6:         Set $\pi_h^k(\cdot \mid s_h) \leftarrow \operatorname{argmax}_{a_h \in \mathcal{A}} Q_h^k(s_h, a_h)$ for all $s_h \in \mathcal{S}$.
 7:         Set $V_h^k(\cdot) \leftarrow \langle \pi_h^k(\cdot \mid \cdot), Q_h^k(\cdot, \cdot) \rangle_{\mathcal{A}}$.
 8:     **end for**
 9:     Obtain $s_1^k$ from the environment.
10:     **for** $h = 1, \dots, H$ **do**
11:         Take $a_h^k \sim \pi_h^k(\cdot \mid s_h^k)$. Obtain $r_h^k = r_h(s_h^k, a_h^k, u_h^k)$ and $s_{h+1}^k$.
12:     **end for**
13: **end for**

---

**Point Estimation.** To solve the Bellman optimality equation in (3.4), we minimize the empirical mean-squared Bellman error as follows at each step,

$$\omega_h^k \leftarrow \operatorname*{argmin}_{\omega \in \mathbb{R}^d} \sum_{\tau=1}^{k-1}\big(r_h^\tau + V_{h+1}^\tau(s_{h+1}^\tau) - \omega^\top \psi_h(s_h^\tau, a_h^\tau)\big)^2 + \lambda\|\omega\|_2^2 + L_h^k(\omega), \ \ h = H, \dots, 1,$$

$$\tag{3.7}$$

where we set $V_{H+1}^k = 0$ for all $k \in [K]$ and $V_{h+1}^\tau$ is defined in Line 7 of Algorithm 1 for all $(\tau, h) \in [K] \times [H - 1]$. Here $k$ is the index of episode, $\lambda > 0$ is a tuning parameter, and $L_h^k$ is a regularizer, which is constructed based on the confounded observational data. More specifically, we define

$$L_h^k(\omega) = \sum_{i=1}^{n}\big(r_h^i + V_{h+1}^k(s_{h+1}^i) - \omega^\top \phi_h(s_h^i, a_h^i, u_h^i)\big)^2, \quad \forall (k, h) \in [K] \times [H], \tag{3.8}$$

which corresponds to the least-squares loss for regressing $r_h^i + V_{h+1}^k(s_{h+1}^i)$ against $\phi_h(s_h^i, a_h^i, u_h^i)$ for all $i \in [n]$. Here $\{(s_h^i, a_h^i, u_h^i, r_h^i)\}_{(i,h) \in [n] \times [H]}$ are the confounded observational data, where

$u_h^i \sim \widetilde{\mathcal{P}}_h(\cdot \,|\, s_h^i)$, $s_{h+1}^i \sim \mathcal{P}_h(\cdot \,|\, s_h^i, a_h^i, u_h^i)$, and $a_h^i \sim \nu_h(\cdot \,|\, s_h^i, w_h^i)$ with $\nu = \{\nu_h\}_{h \in [H]}$ being the behavior policy. Here recall that, with a slight abuse of notation, we write $\mathbb{P}(s_{h+1} \,|\, s_h, a_h, u_h)$ as $\mathcal{P}_h(s_{h+1} \,|\, s_h, a_h, u_h)$ and $\mathbb{P}(u_h \,|\, s_h)$ as $\widetilde{\mathcal{P}}_h(u_h \,|\, s_h)$, since they are induced by the SCM defined in §2.

The update in (3.7) takes the following explicit form,

$$\omega_h^k \leftarrow (\Lambda_h^k)^{-1} \bigg( \sum_{\tau=1}^{k-1} \psi_h(s_h^\tau, a_h^\tau) \cdot \big( V_{h+1}^k(s_{h+1}^\tau) + r_h^\tau \big)$$
$$+ \sum_{i=1}^{n} \phi_h(s_h^i, a_h^i, u_h^i) \cdot \big( V_{h+1}^k(s_{h+1}^i) + r_h^i \big) \bigg), \qquad (3.9)$$

where

$$\Lambda_h^k = \sum_{\tau=1}^{k-1} \psi_h(s_h^\tau, a_h^\tau)\psi_h(s_h^\tau, a_h^\tau)^\top + \sum_{i=1}^{n} \phi_h(s_h^i, a_h^i, u_h^i)\phi_h(s_h^i, a_h^i, u_h^i)^\top + \lambda I. \qquad (3.10)$$

**Uncertainty Quantification.** We now construct the UCB $\Gamma_h^k(\cdot, \cdot)$ of the point estimator $\psi_h(\cdot, \cdot)^\top \omega_h^k$ obtained from (3.9), which encourages the exploration of the less visited state-action pairs. To this end, we employ the following notion of information gain to motivate the UCB,

$$\Gamma_h^k(s_h^k, a_h^k) \propto H(\omega_h^k \,|\, \xi_{k-1}) - H\big(\omega_h^k \,|\, \xi_{k-1} \cup \{(s_h^k, a_h^k)\}\big), \qquad (3.11)$$

where $H(\omega_h^k \,|\, \xi_{k-1})$ is the differential entropy of the random variable $\omega_h^k$ given the data $\xi_{k-1}$. In particular, $\xi_{k-1} = \{(s_h^\tau, a_h^\tau, r_h^\tau)\}_{(\tau,h) \in [k-1] \times [H]} \cup \{(s_h^i, a_h^i, u_h^i, r_h^i)\}_{(i,h) \in [n] \times [H]}$ consists of the confounded observational data and the interventional data up to the $(k-1)$-th episode. However, it is challenging to characterize the distribution of $\omega_h^k$. To this end, we consider a Bayesian counterpart of the confounded MDP, where the prior of $\omega_h^k$ is $N(0, I/\lambda)$ and the residual of the regression problem in (3.7) is $N(0, 1)$. In such a "parallel" confounded MDP, the posterior of $\omega_h^k$ follows $N(\mu_{k,h}, (\Lambda_h^k)^{-1})$, where $\Lambda_h^k$ is defined in (3.10) and $\mu_{k,h}$ coincides with the right-hand side of (3.9). Moreover, it holds for all $(s_h^k, a_h^k) \in \mathcal{S} \times \mathcal{A}$ that

$$H(\omega_h^k \,|\, \xi_{k-1}) = 1/2 \cdot \log \det\big((2\pi e)^d \cdot (\Lambda_h^k)^{-1}\big),$$
$$H\big(\omega_h^k \,|\, \xi_{k-1} \cup \{(s_h^k, a_h^k)\}\big) = 1/2 \cdot \log \det\Big((2\pi e)^d \cdot \big(\Lambda_h^k + \psi_h(s_h^k, a_h^k)\psi_h(s_h^k, a_h^k)^\top\big)^{-1}\Big).$$

Correspondingly, we employ the following UCB, which instantiates (3.11), that is,

$$\Gamma_h^k(s_h^k, a_h^k) = \beta \cdot \Big( \log \det\big(\Lambda_h^k + \psi_h(s_h^k, a_h^k)\psi_h(s_h^k, a_h^k)^\top\big) - \log \det(\Lambda_h^k) \Big)^{1/2} \qquad (3.12)$$

for all $(s_h^k, a_h^k) \in \mathcal{S} \times \mathcal{A}$. Here $\beta > 0$ is a tuning parameter. We highlight that, although the information gain in (3.11) relies on the "parallel" confounded MDP, the UCB in (3.12), which is used in Line 5 of Algorithm 1, does not rely on the Bayesian perspective. Also, our analysis establishes the frequentist regret.

**Regularization with Observational Data: A Bayesian Perspective.** In the "parallel" confounded MDP, it holds that

$$\omega_h^k \sim N(0, I/\lambda), \quad \omega_h^k \,|\, \xi_0 \sim N\big(\mu_{1,h}, (\Lambda_h^1)^{-1}\big), \quad \omega_h^k \,|\, \xi_{k-1} \sim N\big(\mu_{k,h}, (\Lambda_h^k)^{-1}\big),$$

where $\mu_{k,h}$ coincides with the right-hand side of (3.9) and $\mu_{1,h}$ is defined by setting $k = 1$ in $\mu_{k,h}$. Here $\xi_0 = \{(s_h^i, a_h^i, u_h^i, r_h^i)\}_{(i,h) \in [n] \times [H]}$ are the confounded observational data. Hence, the regularizer $L_h^k$ in (3.8) corresponds to using $\omega_h^k \,|\, \xi_0$ as the prior for the Bayesian regression problem given only the interventional data $\xi_{k-1} \setminus \xi_0 = \{(s_h^\tau, a_h^\tau, r_h^\tau)\}_{(\tau,h) \in [k-1] \times [H]}$.

### 3.2 Theory

The following theorem characterizes the regret of DOVI, which is defined in (2.3).

**Theorem 3.5** (Regret of DOVI). Let $\beta = CdH\sqrt{\log(d(T+nH)/\zeta)}$ and $\lambda = 1$, where $C > 0$ and $\zeta \in (0,1]$ are absolute constants. Under Assumptions 3.1 and 3.3, it holds with probability at least $1 - 5\zeta/2$ that

$$\text{Regret}(T) \leq C' \cdot \Delta_H \cdot \sqrt{d^3 H^3 T} \cdot \sqrt{\log(d(T+nH)/\zeta)}, \tag{3.13}$$

where $C' > 0$ is an absolute constant and

$$\Delta_H = \frac{1}{\sqrt{dH^2}} \sum_{h=1}^{H} \big(\log \det(\Lambda_h^{K+1}) - \log \det(\Lambda_h^1)\big)^{1/2}. \tag{3.14}$$

*Proof.* See §F.3 for a detailed proof. $\qquad\square$

Note that $\Lambda_h^{K+1} \preceq (n + K + \lambda)I$ and $\Lambda_h^1 \succeq \lambda I$ for all $h \in [H]$. Hence, it holds that $\Delta_H = \mathcal{O}(\sqrt{\log(n + K + 1)})$ in the worst case. Thus, the regret of DOVI is $\mathcal{O}(\sqrt{d^3 H^3 T})$ up to logarithmic factors, which is optimal in the total number of steps $T$ if we only consider the online setting. However, $\Delta_H$ is possibly much smaller than $\mathcal{O}(\sqrt{\log(n + K + 1)})$, depending on the amount of information carried over by the confounded observational data from the offline setting, which is quantified in the following.

**Interpretation of $\Delta_H$: An Information-Theoretic Perspective.** Let $\omega_h^*$ be the parameter of the globally optimal action-value function $Q_h^*$, which corresponds to $\pi^*$ in (2.3). Recall that we denote by $\xi_0$ and $\xi_K$ the confounded observational data $\{(s_h^i, a_h^i, u_h^i, r_h^i)\}_{(i,h)\in[n]\times[H]}$ and the union $\{(s_h^i, a_h^i, u_h^i, r_h^i)\}_{(i,h)\in[n]\times[H]} \cup \{(s_h^k, a_h^k, r_h^k)\}_{(k,h)\in[K]\times[H]}$ of the confounded observational data and the interventional data up to the $K$-th episode, respectively. We consider the aforementioned Bayesian counterpart of the confounded MDP, where the prior of $\omega_h^*$ is also $N(0, I/\lambda)$. In such a "parallel" confounded MDP, we have

$$\omega_h^* \sim N(0, I/\lambda), \quad \omega_h^* \,|\, \xi_0 \sim N\big(\mu_{1,h}^*, (\Lambda_h^1)^{-1}\big), \quad \omega_h^* \,|\, \xi_K \sim N\big(\mu_{K,h}^*, (\Lambda_h^{K+1})^{-1}\big), \tag{3.15}$$

where

$$\mu_{1,h}^* = (\Lambda_h^1)^{-1} \sum_{i=1}^{n} \phi_h(s_h^i, a_h^i, u_h^i) \cdot \big(V_{h+1}^*(s_{h+1}^i) + r_h^i\big),$$

$$\mu_{K,h}^* = (\Lambda_h^{K+1})^{-1}\bigg(\Lambda_h^1 \mu_{1,h}^* + \sum_{\tau=1}^{K} \psi_h(s_h^\tau, a_h^\tau) \cdot \big(V_{h+1}^*(s_{h+1}^\tau) + r_h^\tau\big)\bigg).$$

It then holds for the right-hand side of (3.14) that

$$1/2 \cdot \log\det(\Lambda_h^{K+1}) - 1/2 \cdot \log\det(\Lambda_h^1) = H(\omega_h^* \,|\, \xi_0) - H(\omega_h^* \,|\, \xi_K). \tag{3.16}$$

The left-hand side of (3.16) characterizes the information gain of intervention in the online setting given the confounded observational data in the offline setting. In other words, if the confounded observational data are sufficiently informative upon the backdoor adjustment, then $\Delta_H$ is small, which implies that the regret is small. More specifically, the matrices $(\Lambda_h^1)^{-1}$ and $(\Lambda_h^{K+1})^{-1}$ defined in (3.10) characterize the ellipsoidal confidence sets given $\xi_0$ and $\xi_K$, respectively. If the confounded observational data are sufficiently informative upon the backdoor adjustment, $\Lambda_h^{K+1}$ is close to $\Lambda_h^1$. To illustrate, let $\{\psi_h(s_h^\tau, a_h^\tau)\}_{(\tau,h)\in[K]\times[H]}$ and $\{\phi_h(s_h^i, a_h^i, u_h^i)\}_{(i,h)\in[n]\times[H]}$ be sampled uniformly at random from the canonical basis $\{e_\ell\}_{\ell\in[d]}$ of $\mathbb{R}^d$. It then holds that $\Lambda_h^{K+1} \approx (K + n)I/d + \lambda I$ and $\Lambda_h^1 \approx nI/d + \lambda I$. Hence, for $\lambda = 1$ and sufficiently large $n$ and $K$, we have $\Delta_H = \mathcal{O}(\sqrt{\log(1 + K/(n+d))}) = \mathcal{O}(\sqrt{K/(n+d)})$. For example, for $n = \Omega(K^2)$, it holds that $\Delta_H = \mathcal{O}(n^{-1/2})$, which implies that the regret of DOVI is $\mathcal{O}(n^{-1/2} \cdot \sqrt{d^3 H^3 T})$. In other words, if the confounded observational data are sufficiently informative upon the backdoor adjustment, the regret of DOVI can be arbitrarily small given a sufficiently large sample size $n$ of the confounded observational data, which is often the case in practice [8, 9, 21, 22, 29].

# 4 Conclusion

In this paper, we propose the deconfounded optimistic value iteration (DOVI) algorithm and its variant DOVI$^+$, which incorporate the confounded observational data to the online reinforcement learning in a provably efficient manner. DOVI and DOVI$^+$ explicitly adjust for the confounding bias in the observational data via the backdoor and frontdoor adjustments, respectively. In both cases, such adjustments allow us to construct the bonus based on a notion of information gain, which considers the amount of information acquired from the offline dataset. We further conduct regret analysis of DOVI and DOVI$^+$. Our analysis suggests that practitioners can tackle the confounding issue in the offline dataset by estimating the counterfactual reward for value function estimations, given that a proper adjustment such as the backdoor or frontdoor adjustment is available. In the case of backdoor and frontdoor adjustment, we prove that the regret of DOVI is smaller than the optimal regret achievable in the pure online setting when the confounded observational data are informative upon the adjustments, suggesting that one can exploit the confounded observational data in reinforcement learning upon proper adjustments. In our future study, we wish to incorporate proxy variables that are native to MDPs for the adjustments of the offline dataset, such as the variables exploited by [4, 24, 40].

## Acknolodgements

Zhaoran Wang acknowledges National Science Foundation (Awards 2048075, 2008827, 2015568, 1934931), Simons Institute (Theory of Reinforcement Learning), Amazon, J.P. Morgan, and Two Sigma for their supports. Zhuoran Yang acknowledges Simons Institute (Theory of Reinforcement Learning). The authors also thank the anonymous reviewers, whose invaluable suggestions help the authors to improve the paper.

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
