# OpenReview forum: "Provably Efficient Causal Reinforcement Learning with Confounded Observational Data"
_NeurIPS.cc/2021/Conference — NeurIPS 2021 Poster_

### Official Review · Reviewer_8cdv · 2021-07-15

**Rating:** 5
**Confidence:** 3

**Summary:**

The paper studies the problem of online reinforcement learning, while making use existing offline data that was previously logged by some behavior policy, where there is unobserved confounding in the logged data. They provide some conditions under which the RL model can be identified from the confounded logged data, and under these conditions propose a variant of optimistic value iteration for online learning that incorporates this offline data as a regularization term, which has an intuitive Bayesian interpretation based on conditioning on this data. Finally, they provide regret bounds for their algorithm, which has a leading term incorporating the effect of the logged data, which decays to zero as the amount of logged data grows to infinity.

**Ethical Concerns:**

I have no ethical concerns.

**Limitations And Societal Impact:**

I have no concerns here.

**Main Review:**

Overall, the paper has many positive aspects; it presents a well-motivated algorithm for an important problem, and provides regret bounds that shrink in terms of the amount of confounded logged data available. In this sense, the paper seems to fill an important niche in the literature. That being said, the paper has several significant weaknesses outlined below, which limits my recommendation. In particular, it feels somewhat lacking/incomplete without any kind of empirical evaluation.

Major weaknesses:
- There is no empirical evaluation of their proposed algorithm, or even an empirical “proof of concept” on synthetic data. This makes it extremely difficult to assess how the algorithm is likely to fair in practice.
- The backdoor criterion in Assumption 3.1 seems to be extremely central to the theory, but it is not explained at all. Ideally, there would be an explanation of when we expect this requirement to be satisfied versus not. Furthermore, it would be very beneficial in motivating the entire paper to provide a compelling clear/concrete example of a problem with unmeasured confounding that fits within your framework and satisfies assumption 3.1. Without such an example, it is difficult to assess how useful the theory is.
- Along a similar vein to the previous concern, the confounding model presented in section 2 seems very weird and non-standard (since, the way it is written, it seems like the confounders are caused by the observed state). This is in contrast to standard kinds of confounding models, where e.g. confounders are independent at each time step, follow in a Markov chain, or follow the POMDP model, etc. Again, it would be great to have a compelling example of a problem with this kind of confounding structure.
- The method is presented as being “provably efficient”. But what does this actually mean? It doesn’t seem to be actually justified as efficient in any concrete sense. One possible sense of efficiency is in terms of semi parametric estimation, but that doesn’t seem to be applied anywhere here. Another sense could be in terms of optimal regret, but this doesn’t seem to be established; although the $\Delta_H$ term in the regret bound has the nice property that it should decay as $n \to \infty$ as discussed in the paper, nothing as far as I can tell rules out the possibility of an alternative algorithm that has faster regret decay as $n \to \infty$. Given this, presenting the work as “provably efficient” in the title is extremely misleading.
- There seems to be a bunch of important recent literature on using confounded off-line RL data that is missing from the related work, for example Namkoong et al. (2020) “Off-policy Policy Evaluation For Sequential Decisions Under Unobserved Confounding”, and Bennett et al. (2021) “Off-policy Evaluation in Infinite-horizon Reinforcement Learning with Latent Confounders”.

Other minor issues / comments / suggestions / typos:
- Typo in line 4 “semple" -> “sample”
- On line 41 you say that “The complicated traffic and poor road design subsequently affect both the action of the drivers and the outcome”. But do we expect these things to be unobserved? This only makes sense as an example of confounding if they are things not observed in the state space, but which the agent took into account. It would be good to elaborate here.
- For fairness, since all presentation and discussion of DOVI+ was deferred to appendix, I am not taking it into account when assessing the paper.
- The discussion of structural causal models at the start of section 2 is very confusingly presented and mostly doesn’t seem to contribute much other than defining the “do” notation (it presents a bunch of notation that is then never used). Also, it would be good to define what the “do calculus” notation means more explicitly for people not already familiar with it.
- The definition of Regret in equation 2.3 defines it as a random quantity (since the definition involves the initial state $s_1^k$ for each $k \in [K]$, which seems somewhat odd. Should there have been an expectation over the initial state?
- You refer to Figure 4 on line 216, but there appears to be no Figure 4 in the paper.
- In a bunch of different places, you talk about replacing  $\mathbb P$ with $\mathcal P_h$. I don’t understand what is meant to be the difference between these two functions; this needs to be explained concretely.
- In assumption 3.3, is $\phi_h$ meant to be a fixed given feature map? This is not explained.
- Also, in assumption 3.3, you refer to the reward function parameters as $\theta_h$, but later in the paper this appears to instead be referred to as $\omega_h$.
- Typo on lines 249-250 “is know” -> “is known”.
- What is the interpretation of the $\beta$ tuning parameter in your algorithm? This doesn’t not appear to be clearly explained. Also, what would be the expected impact of increasing versus decreasing $\beta$ on how the algorithm operates?


**Time Spent Reviewing:**

2

---

> ### Author Response · Authors · 2021-08-10
> **Response to Reviewer 8cdv**
>
> We thank the reviewer for the time dedicated to evaluating our work. We thank the reviewer for the typo spotted. We will revise accordingly. In what follows, we address the questions raised by the reviewer.
>
>
> (1). There is no empirical evaluation of their proposed algorithm, or even an empirical “proof of concept” on synthetic data. This makes it extremely difficult to assess how the algorithm is likely to fair in practice.
>
> We thank the reviewer for suggestions on experiments. In our future revision, we will add experiments to show that utilizing adjustments are necessary for confounded observations.
>
> (2). The backdoor criterion in Assumption 3.1 seems to be extremely central to the theory, but it is not explained at all. Ideally, there would be an explanation of when we expect this requirement to be satisfied versus not. Furthermore, it would be very beneficial in motivating the entire paper to provide a compelling, clear/concrete example of a problem with unmeasured confounding that fits within your framework and satisfies assumption 3.1. Without such an example, it is difficult to assess how useful the theory is.
>
>
> The intuition behind the backdoor is that it summarizes the effect of unobserved confounders on the actions taken. In the sequel, we raise examples that explains both the frontdoor and backdoor criterion.
>
>
> For instance, let us consider a job training recommendation agent that recommends job training programs to clients. The state is the information of clients, and the outcome at one stage is whether the clients obtain a job or not. In the offline data collection, the human job recommender may refuse to recommend programs to certain agents due to, e.g., his personality. Such side information is not recorded as part of the state due to fairness or privacy concerns, and the objective is to train a fair agent that makes recommendations purely based on the agent's skills. In addition, such personality would simultaneously affect finding jobs and form a confounder in the offline dataset. In terms of measuring the effectiveness of job training program recommendations, a natural frontdoor variable is whether the client participates in job training programs. Such event is not affected by the confounders and summarizes all the effectiveness of training program recommendations toward finding jobs. Meanwhile, in the online environment, the autonomous recommender will not be affected by side information since the side information is not included in the states that the autonomous recommender receives. In dynamic treatment, a similar frontdoor is the agent's compliance, that is, whether the patients took the prescribed medication or not.
>
>
> In addition, the intuition behind the backdoor is, in fact, more straightforward than the math may suggest. Broadly speaking, the intuition is to include as many variables available in the offline data as possible, regardless of whether they are part of the states for the trained autonomous agent or not. Upon evaluation with the variables available, in the training of online agents, one should marginalize out the variables that are not considered for the autonomous agent.
>
> For example, in the job training recommendation, if the offline dataset includes information that reflects the agent's personality, such information should be included in the training. Nevertheless, note that our goal is to train agents that make recommendations purely based on the clients' skills. Thus, although the side information is used in fitting rewards for the offline data, they should be marginalized for training the online agents. In addition, in experiments with control groups, being in the control group or not also satisfies the backdoor criterion.
>
> We remark that such a design principle is subtle. Specifically, one must explicitly exclude certain variables, such as the colliders and their descendants, in utilizing the offline data. We refer to, e.g., [1] for more in-depth discussions regarding such an issue.
>
>
>
>
>
>
> (3). Along a similar vein to the previous concern, the confounding model presented in section 2 seems very weird and non-standard (since, the way it is written, it seems like the confounders are caused by the observed state). This is in contrast to standard kinds of confounding models, where e.g. confounders are independent at each time step, follow in a Markov chain, or follow the POMDP model, etc. Again, it would be great to have a compelling example of a problem with this kind of confounding structure.
>
>  We remark that in the offline data generation process, upon conditioning on the state, the relation between the action, the next state and reward, and the confounder form a standard causal analysis model. Specifically, the confounder simultaneously affects both the action and the outcome, namely, the next state and reward. In addition, although our model assumes that the backdoor is affected by the previous state, our analysis still works when it is irrelevant to the states. We refer to (2) for the example.
>
> (4). On provable efficiency.
>
> By claiming the provable efficiency, we refer to the fact that the proposed online learning algorithm has optimal regret. The regret takes the order of $O(\sqrt{T})$, where $T$ is the number of interactions in online learning. In addition to that, we show that the regret is improved if the offline dataset is appropriately adjusted and utilized in online learning. We acknowledge that using "provable efficiency" in our work is more a convention in RL literature and may not be suitable for this work. If necessary, we are willing to modify the title of our work and remove it.
>
> (5). On the missing reference.
>
> We thank the author for pointing out the missing references [2] and [3]. [2] focuses on partial identification under confounding, while [3] uses states and actions as proxy variables to conduct policy evaluation. Both [2] and [3] are highly relevant to our work. We will add a discussion in the intro of our revision.
>
>
>
>
> (6). On the autonomous driving example.
>
> It occurs that the drivers may take actions based on evidence beyond what sensors can capture. For instance, in some cases, the drivers will slow down to avoid being captured by speed sensors that are generally hard to notice. Knowing that there is a speed sensor, in this case, is a domain knowledge that affects the driver's decision-making and is hard for sensors to capture.  We acknowledge that this example is poorly justified and may not be suitable. In our future revision, we will add more discussion on this example and additional examples on decision-making confounders.
>
>
> (7). The discussion of structural causal models at the start of section 2 is very confusingly presented and mostly doesn’t seem to contribute much other than defining the “do” notation (it presents a bunch of notation that is then never used). Also, it would be good to define what the “do calculus” notation means more explicitly for people not already familiar with it.
>
> We thank the reviewer for the suggestions. We will defer notions that are unimportant to the appendix and discuss more on the do calculus in our revision.
>
> (8). The definition of Regret in equation 2.3 defines it as a random quantity (since the definition involves the initial state
> for each $s^k_h$, which seems somewhat odd. Should there have been an expectation over the initial state?
>
> We thank the author for pointing it out. It is missing and is considered a fixed initial state in our analysis. One may wish to consider it a dummy state with $s_2$ being the real stochastic initial state. In our revision, we will remark that $s^k_1$ is fixed throughout the training.
>
> (9). In a bunch of different places, you talk about replacing $\mathbb{P}$ with $\mathcal{P}_h$. I don’t understand what is meant to be the difference between these two functions; this needs to be explained concretely.
>
> In our paper, we use $\mathbb{P}$ to denote the probability. In addition, we use $\mathcal{P_h}$ to denote the transition probability function at each step $h$. There is no difference between $\mathbb{P}(s_{h+1} | s_h, a_h, u_h)$ and $\mathcal{P_h}(s_{h+1} | s_h, a_h, u_h)$. We introduce $\mathcal{P}_h$ as a notation that simplifies the discussion. We will revise our work to clarify our notation.
>
> (10). In assumption 3.3, is meant to be a fixed given feature map? This is not explained.
>
> Yes, in our assumption, we assume that the feature is fixed and is given.
>
> (11). Also, in assumption 3.3, you refer to the reward function parameters as $\theta_h$, but later in the paper this appears to instead be referred to as $\omega_h$.
>
> The parameter $\theta_h$ is the true parameter of the reward mean, whereas the parameter $\omega_h$ is the fitted parameter for Q-functions in our algorithm.
>
> (12). On the tuning parameter $\beta$.
>
> In our algorithm, $\beta$ controls how optimistic the agent is. A large $\beta$ increases the bonus and encourages more exploration but may hinder the exploitation and affect the regret. A small $\beta$ may lead to insufficient exploration and thus affect regret. As shown in our analysis, the appropriate $\beta$ scales with the concentration variance of the target in fitting the Bellman equation.
>
>
> [1] Schölkopf. Elements of Causal Inference. (2017)
>
> [2] Namkoong et al., Off-policy Policy Evaluation For Sequential Decisions Under Unobserved Confounding. (2020)
>
> [3] Bennett et al., Off-policy Evaluation in Infinite-Horizon Reinforcement Learning with Latent Confounders. (2020)

---

> > ### Comment · Reviewer_8cdv · 2021-08-10
> > **Response to Rebuttal**
> >
> > Thanks for the detailed response! I have included a few further clarifications / comments below:
> >
> > - Regarding provable efficiency, I'm very confused. You say in your response that you are talking about having an optimal regret rate with respect to the amount of online data, as per past RL work that proved this optimal regret rate and used the term in that context. However, you literally say in your abstract "...which incorporates the confounded observational data in a provably efficient manner." I still have no idea what you mean by "provably efficient" with respect to the incorporation of the offline data; none of the results you presented imply that your regret is rate-optimal in terms of the usage of this data in any sense. Given that, this is highly misleading.
> > - I had a skim over the paper again and the confounder model makes more sense to me now, so ignore my comment about it, it's clear to me now that it's a special case of the MDPUC model.
> > - The job training example you gave doesn't seem very appropriate, since there is no dynamic element (this seems more like a contextual bandit problem). It would be much more appropriate to have an actual dynamic RL-type example.
> > - Thanks for the detailed discussion about the backdoor criterion. However, my question was more motivated by making things easier to understand for readers more familiar with the potential outcomes style of causal inference (which is very common for researchers studying offline policy evaluation) rather than the Pearl style. For such readers, talking about complex graph-theoretic concepts like "d-separation" without explanation could be very confusing. My intuitive understanding is that we require that the observed action and potential rewards (i.e. values of $r(s_h,a,w_h)$ for each value of $a$, or equivalently the rewards that would have been obtained if, possibly counter to fact, action $a$ were taken) are conditionally independent given both $u_h$ and $s_h$ (i.e. that we satisfy an "ignorability" kind of condition given both $s_h$ and $u_h$), is this correct?
> > - Also, a follow up question about the backdoor criterion after thinking about it more, why is there only an assumed backdoor criterion for the effect of $a_h$ on $s_{h+1}$, and not for the effect of $a_h$ on $r_h$?

---

> > > ### Author Response · Authors · 2021-08-11
> > > **Response to Reviewer 8cdv**
> > >
> > > We thank the reviewer again for taking the time to respond to us and give insightful discussions. In the sequel, we respond to the questions raised.
> > >
> > > (1). We remark that in our work, the efficiency does not refer to the fact that the estimator has a variance that matches the Cramer-Rao lower bound (e.g., as in semiparametric efficiency). In reinforcement learning (RL), the term "efficiency" is an abbreviation of sample efficiency. The term describes the fact that the regret in online learning has a regret that scales with $o(T)$, where $T$ is the total number of interactions with the environment. Similar claims also arise in the previous study of RL theory (see, e.g., [1][2][3]).
> > >
> > > In our work, the proposed online learning algorithm has a regret that takes the order $O(\sqrt{T})$, where $T$ is the number of interactions in online learning. Moreover, we show that the regret is improved if the offline dataset is appropriately adjusted and utilized in online learning, which is why we make the claim "...which incorporates the confounded observational data in a provably efficient manner."
> > >
> > >
> > >
> > >
> > >
> > >
> > > (2). We remark that the job training recommendation task is a sequential task. The clients that do not find jobs have the chance to be recommended recursively. In such a scenario, the learning fits into our proposed confounded MDP framework.
> > >
> > >
> > > (3). Yes, indeed, it is possible to relate the backdoor framework to ignorability.  For the dynamics with only single-step transition form action $A$ to outcome $R$, having a variable $U$ that satisfies the backdoor criterion means that the treatment assignment is ignorable, that is, $R(a) \perp A | U$. It is also possible to extend the description to the sequential decision-making problem that we consider. In our revision, we will add some discussion on the description with ignorability.
> > >
> > > (4). Thanks for pointing out the issue. It is, in fact, a typo: we missed the assumption that the backdoor criterion holds for the effect of $a_h$ on $r_h$. In fact, we use such an assumption to obtain Proposition 3.2. In our revision, we will add the assumption into Assumption 3.1.
> > >
> > >
> > >
> > >
> > >
> > > [1] Fei et al., Provably Efficient Exploration for RL with Unsupervised Learning. (2020)
> > >
> > > [2] Du et al., Provably Efficient Q-learning with Function Approximation via Distribution Shift Error Checking Oracle. (2019)
> > >
> > > [3] Wang et al., Reinforcement Learning with General Value Function Approximation: Provably Efficient Approach via Bounded Eluder Dimension. (2020)

---

> > > > ### Author Response · Authors · 2021-08-18
> > > > **Additional concerns?**
> > > >
> > > > Dear reviewer 8cdv,
> > > >
> > > > Thanks so much for the time and effort you spent reviewing our work! Please let us know if our response addressed your concerns. If further clarifications are necessary, we are willing to provide them.

---

### Official Review · Reviewer_uXJG · 2021-07-16

**Rating:** 6
**Confidence:** 2

**Summary:**

This is a technical paper presenting how to incorporate offline observational data to improve the sample efficiency in the online reinforcement learning setting.
The issue is the potential presence of unobserved confounders in the observational data which impact the transition dynamics and the rewards and how to adjust the exploration bonus used in the online setting.
The authors suggest an algorithm (DOVI) which adjusts for the confounding bias (where the coufounders are partially observed or unobserved). They then derive a bound on the regret when using linear function approximation which shows that the regret is smaller than the optimal online regret thanks to the use of offline observational data if they are informative.

**Limitations And Societal Impact:**

-

**Main Review:**

The paper is well-written and clear. There are no experiments. While this is not particularly an issue for this kind of technical paper, a toy example illustrating the advantage of using offline observational data and the different regimes of the regret bound would have been beneficial.

The paper is well-motivated: it is very often the case that observational data are available and it is indeed relevant to try to use this data to reduce the sample cost of existing deep reinforcement learning solutions, especially when simulators are not available as is the case for most real-life problems (engineering systems, health, ....). Leveraging tools from causal inference is an interesting direction that is increasingly popular. I believe the theoretical result to be interesting for the community.

Major comments:
- the definition of an intervention states that the value is assigned regardless of the other exogenous and endogenous variables. However in the online context, when using a policy, the authors uses the do operator (as an intervention) but the policy selects an action based on the state $s$. This is confusing as the action is thus not assigned regardless of the state which is an endogenous variable.
- Are the authors the first to state a Backdoor Adjusted Bellman Equation? this is not clear in the paper.
- It is also not clear to me where the distinction between the observational and interventional data is in algorithm 1?

Minor comments:
- the abstract format is not correct.
- line 4: semple -> sample
- lines 56 and 57: please give references.

**Time Spent Reviewing:**

4

---

> ### Author Response · Authors · 2021-08-10
> **Respose to Reviewer uXJG**
>
> We thank the reviewer for the time dedicated to evaluating our work. We thank the reviewer for the typo spotted and the lack of reference. We will revise accordingly. In what follows, we address the questions raised by the reviewer.
>
>
> (1). The definition of an intervention states that the value is assigned regardless of the other exogenous and endogenous variables. However in the online context, when using a policy, the authors use the do operator (as an intervention) but the policy selects an action based on the state. This is confusing as the action is thus not assigned regardless of the state which is an endogenous variable.
>
>
> In our work, we interpret the policy in online learning as the stochastic intervention, which generalizes the notion of intervention and enforces the action to be a random variable upon conditioning on base variables. Such intervention is also studied in previous analyses of treatment effects (e.g., [1][2]).
>
> (2). Are the authors the first to state a Backdoor Adjusted Bellman Equation? this is not clear in the paper.
>
> A similar notion arises in causal bandits reviewed in section B of our appendix. In addition, the sequential backdoor is previously studied by [3], which aims to compute the probability after intervention in a sequential decision-making problem. Nevertheless, as far as we are concerned, we are the first to propose such an adjusted Bellman equation in the sequential decision-making problem.
>
>
> (3). It is also not clear to me where the distinction between the observational and interventional data is in algorithm 1?
>
> For the observation data, the decisions are affected by unobserved confounders, making it challenging to infer transition based on observation data. The DAG that describes the observational data generation process is in Fig 1(a). Interventional data are collected in the online learning process, where the policy depends solely on states. The DAG that describes the observational data generation process is in Fig 1(b).
>
>
> [1] Dıaz and Hejazi, Causal mediation analysis for stochastic interventions. (2020)
>
> [2] Díaz and van der Laan, Population Intervention Causal Effects Based on Stochastic Interventions. (2012)
>
> [3] Pearl and Robins, Probabilistic evaluation of sequential plans from causal models with hidden variables. (1995)

---

> > ### Comment · Reviewer_uXJG · 2021-08-23
> > **Thank you**
> >
> > Thank you for the detailed answers to my comments.

---

### Official Review · Reviewer_czeu · 2021-07-17

**Rating:** 6
**Confidence:** 2

**Summary:**

This paper proposed an algorithm to learn optimal policy reinforcement learning in the presence of confounded observational data. It proposes to remove confounders from the data and improve sample efficiency in online settings. It addresses two scenarios, partially observed and unobserved confounder, by using two techniques, the frontdoor, and backdoor criterion. It further theoretically analyzes the regret bound in these settings.


**Limitations And Societal Impact:**

The assumptions are stated clearly in the papers. However, additional discussion about the feasibility of these assumptions needs to be discussed (as suggested in detailed comments).


**Main Review:**

Originality:  This paper addresses an important problem of leveraging confounded observational data in learning optimal policy in the reinforcement learning framework. The work is novel, and it incorporates some well-known techniques (Frontdoor, backdoor adjustment) in the context of reinforcement learning. Related works need to be included in how these works fall under the existing (deep) reinforcement learning framework.

Quality: The work is technically sound,while an empirical evaluation is needed to confirm the effectiveness of the proposed algorithms


Clarity: The paper is well-written and easy to follow. Necessary assumptions are stated.

Significance: The work has value and has the potential to be used by others. However, there are further discussions needed on how the works fit into the existing reinforcement learning literature.


Detailed comments:

The proposed algorithms and bound is analyzed based on the linearity assumptions (i.e., linear SCM, linear confounded MDP). How is the analysis dependent on this assumption? How can it be generalized to non-linear cases? If it is not straightforward, I suggest explicitly mentioning this assumption in the abstract and introduction.

The paper motivates the neural network function approximation in deep reinforcement learning. However, it is still unclear to me if such a scenario is discussed in the context of the proposed method.

Is Algorithm 1 assumed a tabular reinforcement learning setup? If that is the case, how those bounds hold with function approximation, such as using a neural network to approximate value function, Q-function, and policy. These are the setups used extensively in current RL literature (deep reinforcement learning).

In Algorithm 1, how the online data is used to estimate Value, Q-function, and policy? It seems the online data line 10-12 is not saved in any buffer. How does it then influence the policy?

How do these assumptions, frontdoor exist, be mapped from the real-world example?

How are these methods relevant to deep reinforcement learning, as is motivated in the abstract and intro? Need some discussion on this line, though the author mentions this is relevant to causal bandits.




How partially observed MDP is handled, that is, backdoor criterion assumption limits the use of the proposed algorithms. What are some practical scenarios that hold this assumption in the context of reinforcement learning? Justifying this would be useful for readers.
How to derive a practical RL algorithm?

Backdoor adjustment limits the applicability of the algorithm/method. Can you give some practical scenarios where these are held?

What if the confounder is fully observed? What difference it will make compared to partially observed. How does it impact the regret bound or algorithmic convergence?

Not clear why the reward function in equation 3.2 is called the counterfactual reward function.



Is the observational data (first line of Algorithm 1) collected in an offline setting? How the online data (line 10-12 in Algorithm 1) is used? Is it stored in a buffer and then merged with the observational data? What policy (random or human demonstration) was used to collect the observational data?



Empirical evaluation will strengthen the papers and verify the feasibility of the proposed algorithm in practice. While I understand this paper focuses on theory, the feasibility of the assumptions needs to be discussed in the paper. For example, an example can be added with stating the assumptions of how they relate to some real-world settings, which will help readers better understand the implication of the proposed algorithms.

The literature review should address how this paper’s algorithm fits into existing RL literature, which seems to be missing in the paper. Section B discusses the comparison with causal bandits.

How does the confounded MDP relate to the previously proposed Block MDP[1]?

The observational data requires an additional record of confounders (partially observed confounders and intermediate state) which standard policy in the RL framework does not provide. So how do the observational data need to be collected to be incorporated in the online setup? Need to discuss in the paper.

The paper seems to end abruptly without a proper conclusion. I suggest adding a conclusion or discussion section summarizing the contribution and results.



Reference:
[1] Provably efficient rl with rich observations via latent state decoding. Simon Du, Akshay Krishnamurthy, Nan Jiang, Alekh Agarwal, Miroslav Dudik, and John Langford. In International Conference on Machine Learning, pages 1665–1674. PMLR, 2019.



**Time Spent Reviewing:**

6 hours

---

> ### Author Response · Authors · 2021-08-10
> **Response to Reviewer czeu**
>
> We thank the reviewer for the time dedicated to evaluating our work. In what follows, we summarize the concerns raised by the reviewer and address them respectively.
>
> (1). On generalizing linear models to general function approximations.
>
> Generalizing our work to the non-linear models is not straightforward. It is a challenging problem even in the ordinary MDP.  Based on the recent advances in RL theory with general function approximation (e.g., [1][2][3][4]), it is possible to extend our analysis based on NTK theory or general function approximation with the eluder dimension or Bellman rank. The key of our linear assumption is to ensure that, upon taking the Bellman operator, the value functions (with added bonus) still fall in the function parameterization space that we consider. Such property is sometimes known as Bellman complete and has been extensively studied in the literature (e.g., [5],[6],[7]). In general, assuming Bellman complete or low inherent Bellman error is a weaker assumption and is a possible direction to extend our work. The function approximation is not the focus of this work but is an important future direction of our future study. In our revision, we will add the discussion on generalizing to general function approximations.
>
>
>
>
>
>
> (2). On deep learning instantiation.
>
> (i). In our paper, we focus on linear setup. Our method does not restrict the form of feature embedding, which could itself be a neural network. In fact, for DQN typed learning algorithms, the output layer of Q-functions can be a linear layer. With such function parameterization, one can utilize the fitted feature embedding function for exploration (e.g., [8][9]).
>
>
> (ii). More importantly, the key observation of our analysis is a design principle of RL with confounding. Specifically, our work shows that if the causal effect can be identified, one can compute the counterfactual of value functions based on the offline dataset and use it as the target in fitting the Bellman equation. Specifically, in our example, when the data $(s, a, r, u)$ is available, the correct target one needs to use is the following counterfactual reward,
> $$
> \mathbb{E}_u[r(s, a, u) + \gamma * V(s')].
> $$
> Here $r(s, a, u)$ is the fitted reward with $(s, a, u)$ as the covariate, and the expectation is taking with respect to the marginal of $u$, which can be estimated by sampling $u$ from the buffer.
>
> (3). On the setup of our problem.
>
> Our method considers feature embedding with a general state space and is not limited to the tabular setting. Our regret only depends on the dimension of the feature and is irrelevant to the state space. Though our theory is restricted to linear models, it is possible to generalize our theory to neural networks based on recent advances in RL theory, as discussed in (1).
>
>
> (4). On the collection and usage of online data in the algorithm.
> The offline data are collected by a behavior policy in the offline environment in Figure 1(a), and the online data are collected as in Line 11 of the algorithm.
>
> Both the data collected are used to construct loss function and bonus in equations (3.8) and (3.12), respectively. The data influences the policy through both the bonus and the parameter of Q-functions.
>
> (5). On the real-world example of backdoor and frontdoor criteria.
>
>
> For instance, let us consider a job training recommendation agent that recommends job training programs to clients iteratively. The state is the information of clients, and the outcome at one stage is whether the clients obtain a job or not. In the offline data collection, the human job recommender may refuse to recommend programs to certain agents due to, e.g., his personality. Such side information is not recorded as part of the state due to fairness or privacy concerns, and the objective is to train a fair agent that makes recommendations purely based on the agent's skills. In addition, such personality would simultaneously affect finding jobs and form a confounder in the offline dataset. In terms of measuring the effectiveness of job training program recommendations, a natural frontdoor variable is whether the client participates in job training programs. Such event is not affected by the confounders and summarizes all the effectiveness of training program recommendations toward finding jobs. Meanwhile, in the online environment, the autonomous recommender will not be affected by side information since the side information is not included in the states that the autonomous recommender receives. In dynamic treatment, a similar frontdoor is the agent's compliance, that is, whether the patients took the prescribed medication or not.
>
>
> In addition, the intuition behind the backdoor is, in fact, more straightforward than the math may suggest. Broadly speaking, the intuition is to include as many variables available in the offline data as possible, regardless of whether they are part of the states for the trained autonomous agent or not. Upon evaluation with the variables available, in the training of online agents, one should marginalize out the variables that are not considered for the autonomous agent.
>
>
> For example, in the job training recommendation, if the offline dataset includes information that reflects the agent's personality, such information should be included in the training. Nevertheless, note that our goal is to train agents that make recommendations purely based on the clients' skills. Thus, although the side information is used in fitting rewards for the offline data, they should be marginalized for training the online agents. In addition, in experiments with control groups, being in the control group or not also satisfies the backdoor criterion.
>
> We remark that such a design principle is subtle. Specifically, one must explicitly exclude certain variables, such as the colliders and their descendants, in utilizing the offline data. We refer to, e.g., [10] for more in-depth discussions regarding such an issue.
>
>
>
>
>
> (6) On the name "counterfactual reward".
>
> We call it counterfactual as it is the reward if we adopt do operation on the action. That said, the counterfactual reward is the expected reward when action depends solely on the previous state. Note that the actions of the offline data collection process depend on an unobserved confounder. The notion of counterfactual reward arises as it is the reward that we need to estimate to obtain the optimal policy.
>
> (7). On the ending of the paper.
>
> We thank the reviewer for the suggestion of adding a conclusion. In our revision, we will add a section to summarize our theoretical findings and discuss the deep learning instantiation and real-world applications raised in (2) and (5), respectively.
>
>
>
> [1] Russo and Van Roy, Learning to Optimize Via Posterior Sampling. (2014)
>
> [2] Ayoub et al., Model-Based Reinforcement Learning with Value-Targeted Regression. (2020)
>
> [3] Yang et al., On Function Approximation in Reinforcement Learning: Optimism in the Face of Large State Spaces. (2020)
>
> [4] Jiang et al., Contextual Decision Processes with Low Bellman Rank are PAC-Learnable. (2016)
>
> [5] Du et al., Bilinear Classes: A Structural Framework for Provable Generalization in RL. (2021)
>
> [6] Zanette et al., Learning Near Optimal Policies with Low Inherent Bellman Error. (2020)
>
> [7] Xie et al., Bellman-consistent Pessimism for Offline Reinforcement Learning. (2021)
>
> [8] Azizzadenesheli and Anandkumar, Efficient Exploration through Bayesian Deep Q-Networks. (2019)
>
> [9] O'Donoghue et al., The Uncertainty Bellman Equation and Exploration. (2018)
>
> [10] Schölkopf. Elements of Causal Inference. (2017)

---

### Official Review · Reviewer_wam3 · 2021-07-20

**Rating:** 6
**Confidence:** 4

**Summary:**

In the main paper, the authors study the problem of performing value iteration using confounded observational data where the confounders are partially observable. Applying for backdoor adjustment to correct the observational data, the authors propose de-confounded optimistic value iteration (DOVI) in this setting. Finally, a sublinear regret bound of DOVI is provided for linear confounded MDPs and cases where the observed subset of the confounders satisfy the backdoor criterion.

**Limitations And Societal Impact:**

Based on the checklist answer, the authors did not address the limitations of their work.

**Main Review:**

The paper proposes an interesting problem to study and has a clear motivation. DOVI is an adaptation of least squares value iteration in linear confounded MDPs. The authors have given a clear description of the algorithm and its guarantee.

(1) Can the authors comment on whether DOVI can be considered an instantiation of https://arxiv.org/pdf/2011.04622.pdf with a particular feature map and regularization? Given that we only observe samples of s_h, u_h, how should one obtain the back-door adjusted feature in general? And how will the estimation error of the back-door adjusted feature play a role in the regret?

(2) Experiments:
- It would be very helpful to showcase how DOVI performs empirically compared to VI methods that do not take confounding into account, i.e., treating u_h as part of the state variable.
- Since the regret guarantee greatly depends on the linear confounded MDPs assumption, performing some sensitivity analysis on that would be very important.

Overall, the strengths of the paper are (1) a clean formulation of confounded MDPs and (2) DOVI which achieves sublinear regret for linear confounded MDPs. The main weaknesses of the paper are (1) (theoretical and empirical) comparison of DOVI with other algorithms in tackling confounded MDPs (e.g., general algorithms for solving POMDPs or algorithms that are designed for unconfounded MDPs); (2) a clearer description on how DOVI is related to existing least-squares based VI algorithms.

Minor typo:
- L4: "semple" -> "sample"

*Update after the author response*
I have read through the author response and other reviews. The author response has clarified some of my concerns. Similar to reviewer czeu and 8cdv, I believe that an adequate experiment section is needed for the paper. Hence, I am maintaining my score.

**Time Spent Reviewing:**

2 hours

---

> ### Author Response · Authors · 2021-08-10
> **Response to Reviewer wam3**
>
> We thank the reviewer for the time dedicated to evaluating our work. We thank the reviewer for the typo spotted. We will revise accordingly. In what follows, we address the questions raised by the reviewer.
>
> (1). Our work cannot be considered as an instantiation of [1]. Our work aims to utilize the confounded offline data for online learning, whereas [1] considers the online exploration problem. [1] aims to propose efficient exploration algorithms under general function approximations, which is related to our work but has a different focus. In terms of the feature construction, our work shows that we need to utilize different embedding for the offline and online dataset, respectively, whereas [1] has only online data and does not consider such a difference in embeddings. The backdoor adjusted feature is obtained by taking expectations with respect to $u$ over the feature embedding of $(s, a, u)$. Empirically, such a feature can be obtained by embedding offline and online datasets with distinct embeddings and then minimizing the deviation between offline data embedding upon taking the empirical mean over $u$ and the online data embedding. The estimation error of features introduces an additional model misspecification error in the transition models, which subsequently affects the estimation of Q-functions. Such error further affects the regret and results in a linear error in the estimation error of features.
>
>
> (2). (i) In our future revision, we will try to add experimental results demonstrating that ignoring the confounder will lead to biased estimates.
>
> (ii) Both VI and our method utilizes proxy variables to identify the causal effect. As a comparison, VI utilizes instrument variables, whereas our analysis utilizes frontdoor and backdoor criteria.
>
> (iii) Our analysis raises the assumptions on DAG, which allows identification of the causal effects by the backdoor and frontdoor criteria. When such assumptions fail to hold, the problem has only partial identification, and sensitivity analysis is necessary. The major difference in such analysis compared to our work is that it only makes sense to compare against the best agent with the exact partial identification ability of the model in the partial identification setting. Equivalently, the definition of regret in our work is no longer applicable as it is impossible to be comparable with the best agent that knows the exact transition. Like the robust MDP setting, a possible alternative is to compare under the worst MDP possible within the possible models upon partial identification. Such analysis is challenging and is an important future direction of our research.
>
> [1] Yang et al., On Function Approximation in Reinforcement Learning: Optimism in the Face of Large State Spaces. (2020)

---

### Decision · Program_Chairs · 2021-09-27

**Decision:**

Accept (Poster)

**Comment:**

Reviewers were mostly agreed that the paper makes an important contribution at the interface of RL and causal inference, proposing new ways of using confounded offline data to help online RL and rigorously proving their value. Thus, the paper brings together familiar tools from both fields (causality and RL) in a fruitful way that could serve as a basis for future work.
The major weakness of the paper is lack of any experimental validation. While the theoretical contribution is sufficiently important in itself, adding even a synthetic experiment would make the paper stronger.
Finally, the reviewers pointed out several aspects where the paper could be made clearer, and I urge the authors to address these issues in their final version.